# Validation of the Reference Genes for Expression Analysis in the Hippocampus after Transient Ischemia/Reperfusion Injury in Gerbil Brain

**DOI:** 10.3390/ijms24032756

**Published:** 2023-02-01

**Authors:** Anita Lewczuk, Anna Boratyńska-Jasińska, Barbara Zabłocka

**Affiliations:** Mossakowski Medical Research Institute, Polish Academy of Science, 02-106 Warsaw, Poland

**Keywords:** brain ischemia, hippocampus, mRNA expression, reference genes

## Abstract

Transient brain ischemia in gerbils is a common model to study the mechanisms of neuronal changes in the hippocampus. In cornu ammonnis 2–3, dentate gyrus (CA2-3,DG) regions of the hippocampus, neurons are resistant to 5-min ischemia/reperfusion (I/R) insult, while cornu ammonnis 1 (CA1) is found to be I/R-vulnerable. The quantitative polymerase chain reaction (qRT-PCR) is widely used to study the expression of genes involved in these phenomena. It requires stable and reliable genes for normalization, which is crucial for comparable and reproducible analyses of expression changes of the genes of interest. The aim of this study was to determine the best housekeeping gene for the I/R gerbil model in two parts of the hippocampus in controls and at 3, 48, and 72 h after recanalization. We selected and tested six reference genes frequently used in central nervous system studies: *Gapdh*, *Actb*, *18S rRNA*, *Hprt1*, *Hmbs*, *Ywhaz*, and additionally *Bud23*, using RefFinder, a comprehensive tool based on four commonly used algorithms: delta cycle threshold (Ct), BestKeeper, NormFinder, and geNorm, while *Hprt1* and *Hmbs* were the most stable ones in CA2-3,DG. *Hmbs* was the most stable in the whole hippocampal formation. This indicates that the general use of *Hmbs*, especially in combination with *Gapdh*, a highly expressed reference gene, seems to be suitable for qRT-PCR normalization in all hippocampal regions in this model.

## 1. Introduction

The search for mechanisms of neuronal death and regeneration in rodent models of stroke or brain ischemia and reperfusion episodes is not only very important for our understanding of the phenomenon but mainly for the identification of future therapeutic targets. Regardless of the type of neurons (pyramidal in cornu ammonnis 1, 2 or 3 (CA1, CA2, CA3) or granule cells in dentate gyrus (DG)) only the CA1 region undergoes delayed neuronal death, while the resistance to ischemia/reperfusion (I/R) insult is observed in the other hippocampal regions; however, the molecular mechanisms responsible for this phenomenon are not fully elucidated and are still under investigation [1,2]. Reverse transcription followed by quantitative polymerase chain reaction (qRT-PCR) is a powerful and commonly used tool for gene expression analysis. However, normalization of the data obtained requires stable reference genes throughout time and experimental conditions in the brain tissue, as these genes are crucial for reliable and reproducible analysis of gene expression [3,4,5,6,7]. Another approach to validation of gene expression results was reported in the paper by Kovac et al., 2016 [8], where electrophoretic gels and linear regression plots of qRT-PCR were presented for target GABAAR α1 subunit mRNA and the associated internal standard for sham and lesioned animals. Today, reference gene expression is precisely determined for a given species, tissue, and experimental model, taking into account genes coding for proteins involved in cell metabolism, so-called housekeeping genes [9,10,11,12].

Transient middle cerebral artery occlusion (tMCAO) was used to test *Gapdh*, *Actb*, *Hprt1*, *Hmbs*, and *Gusb* as an internal standard for normalization of gene expression data in mice brains of 2-, 9-, 15-, and 24-month-old animals, 2 and 7 days after reperfusion [13]. The use of *Gapdh* and *Actb* as stable and highly abundant transcripts was evaluated for the normalization of qRT-PCR data. However, for low-abundance genes, the use of *Hmbs* in the brain is recommended [13]. After 90 min of tMCAO and 12 and 24 h or 3 and 7 days of reperfusion in three areas of rats’ brains (frontal cortex, hippocampus, and dorsal striatum), *Ppia* appeared as the best reference gene, with *Ywhaz* and *Hprt1* as complements for multigene normalization [14]. In 10-min global brain ischemia in rats due to cardiac arrest followed by reperfusion for up to 30 days, the housekeeping gene *Rpl13a* was used as an internal control gene [15]. Several candidate reference genes were evaluated in permanent middle cerebral artery occlusion (pMCAO) in rats [16]. *Sdha* and *Ywhaz* were suggested as appropriate for this in vivo model. Nevertheless, genes such as *Gapdh* or *Actb*, traditionally used as reference in in vivo and in vitro ischemia and reperfusion models, are still in use [17]. However, it was reported that the expression levels of these two genes were affected by ischemia and were not among the top ten candidates for control genes in mouse and rat global cerebral ischemia and reperfusion models [18,19]. Although the combination of various genes is recommended for various stroke models [20,21] it is important to validate and select a proper gene according to the experimental conditions and species of animals.

The present study was initiated to determine the stability of seven genes (*Gapdh*, *Actb*, *18S rRNA*, *Hprt1*, *Hmbs*, *Ywhaz*, and *Bud23*) using qRT-PCR to establish their suitability as housekeeping genes for an in vivo cerebral ischemia and reperfusion model in gerbils. After 5 min of common carotid occlusion, postischemic neuronal death develops gradually after reperfusion and selectively damages neurons in specific areas of the brain due to their specific intrinsic vulnerability. In rodent models, global ischemia–reperfusion injury typically affects neurons in the hippocampal CA1 area, while neurons in the CA2–CA3 areas and dentate gyrus granule cells are reported to be relatively resistant [22,23,24].

Although *Gapdh*, *Actb*, *18S rRNA*, *Hprt1*, *Hmbs*, and *Ywhaz* are widely used as housekeeping genes in a context of reference gene expression, the *Bud23* gene was also selected in this study. This gene encodes rRNA methyltransferase and ribosome maturation factor, the protein that specifically methylates the N_7_ position of a guanine in *18S rRNA*. It is involved in ribosome biogenesis, especially pre-rRNA processing steps leading to small subunit rRNA production independently of its RNA-modifying catalytic activity [25]. The methyl transferase activity of Bud23 is dispensable for cell growth; however, *Bud23* deletion leads to a strong defect in subunit export. It is suggested that Bud23 promotes a critical transition event to facilitate folding of the central pseudoknot of the small subunit, so it serves as a late checkpoint for monitoring the export competence of the small subunit of rRNA [25,26]. Recently, a mechanism that regulated mitochondrial content and function through the generation of Bud23-dependent ribosomes was recorded in cardiac cells [27]. A tumor-promoting role of *Bud23* in glioma cells [28] and its function as a regulator of chromatin structure were also reported [29].

The aim of this study was to evaluate the best housekeeping gene for the I/R gerbil model in selected parts of the hippocampus at different time points after recanalization. Previous reports were focused on the mechanisms of delayed neuronal death or the explanation of natural adaptive mechanisms (called endogenous neuroprotection). Resistance to ischemia–reperfusion injury in the CA2-3,DG region of the hippocampus versus the dorsal, ischemia-vulnerable region (CA1) in the gerbil is a well-known example of this phenomenon. Considering our previous data on ischemia-induced molecular cascades, in this study we selected three time points: 3, 48, and 72 h. Reportedly, the first three days after the restoration of circulation are the most important for the transduction of intracellular ischemic signals and molecular changes that result in delayed neuronal death or survival, as studied in animal models and in clinical practice [2]. Here, the following genes were evaluated: *Gapdh*, *Actb*, *18S rRNA*, *Hprt1*, *Hmbs*, *Ywhaz*, and *Bud23* in the RefFinder algorithm, which combines four commonly used computational programs: delta cycle threshold (Ct), BestKeeper, NormFinder, and geNorm to find the best reference gene in order to identify changes in gene expression in a gerbil model of transient cerebral ischemia so that we could gain a more in-depth understanding of the molecular mechanisms behind the differential sensitivity of hippocampal regions to an ischemic episode.

## 2. Results

### 2.1. qRT-PCR Primer Efficiency and Specificity and Expression Levels of Candidate Reference Genes

The analysis of standard curves resulted in an amplification efficiency ranging from 90.10% (*Hprt1*) to 107.05% (*Ywhaz*), while the regression coefficient ranged from 0.997 (*Ywhaz*) to 0.999 (*Gapdh*, *Actb*, *18S rRNA*, *Hprt1*, *Hmbs*), whereas for *Bud23* it reached a value of 0.998 (Table 1). Primer specificity was assessed by a melting curve analysis that showed a single peak of melting curve for all seven primer pairs (Figure 1), indicating the absence of nonspecific polymerase chain reaction (PCR) amplification. The distribution of cycle threshold (Ct) values revealed the lowest mean Ct value of 12.28 for *18S rRNA*, while the highest mean Ct value was observed for *Hmbs* (30.23) (Figure 2, Appendix A). The lowest Ct range (2.36 cycles) was detected for *Hmbs*, while the greatest discrepancy in Ct values (4.53) was observed for *18S rRNA*, which is reflected in their standard deviation (SD) scores (Appendix A). The *18S rRNA* gene had the highest level, while *Hmbs* and *Bud23* had the lowest levels of mRNA transcript among the genes examined in the gerbil hippocampus samples.

### 2.2. Comprehensive Ranking of Candidate Reference Genes in CA1 and CA2-3,DG of Gerbil Hippocampus

Four algorithms were used: delta Ct, BestKeeper, NormFinder, and geNorm, jointly analyzed in the RefFinder [32], to establish the most stable reference gene in the gerbil model of transient brain ischemia. Expression analysis was performed in controls and animals subjected to ischemia and 3, 48, or 72 h of reperfusion in CA1 and CA2-3,DG hippocampal sections.

#### 2.2.1. ΔCt Method

The comparison of average standard deviation made by the ΔCt method indicated that *Gapdh*, *Hmbs*, *Actb*, and *Bud23* were the most stably expressed in the CA1 sector during postischemic reperfusion (Figure 3A1), whereas *Hprt1*, *Hmbs*, *Bud23*, and *Gapdh* were the most stably expressed in the CA2-3,DG sector (Figure 3A2), with the SD average below 0.6 for each gene (Appendix A). Slightly higher values, 0.61 and 0.62, were recorded for *Ywhaz* in CA1 and CA2-3,DG, respectively. *Hprt1* and *18S rRNA* showed the highest fluctuation of ΔCt in CA1, and *18S rRNA* and *Actb* showed the highest fluctuation in CA2-3,DG.

#### 2.2.2. BestKeeper

The BestKeeper algorithm indicated that *Hmbs* and *Actb* were the most stable reference genes in CA1, with a standard deviation of 0.363 and 0.436, respectively (Figure 3B1, Appendix A). They were followed by *Bud23*, *Gapdh*, and *Hprt1*, with similar results—nearly 0.56 each. The least stable reference genes in CA1, according to the BestKeeper algorithm, were observed in *Ywhaz* (0.673) and *18S rRNA* (1.102). BestKeeper analysis revealed that *Hprt1* (0.416) was the most stable gene for CA2-3,DG (Figure 3B2, Appendix A). The following genes: *Ywhaz*, *Hmbs*, *Bud23*, and *Gapdh* showed a result below 0.56. *Actb* and *18S rRNA* were the least stable genes in CA2-3,DG.

#### 2.2.3. NormFinder

Based on NormFinder results, the genes that were expressed most stably included *Gapdh* (stability value 0.177) and *Hmbs* (stability value 0.241) in the CA1 and CA2-3,DG sectors, respectively (Figure 3C1,C2, Appendix A). In the CA1 sector, the order of the remaining genes was as follows: *Hmbs* > *Actb* > *Ywhaz* > *Bud23* > *Hprt1* > *18S rRNA*, and in CA2-3,DG: *Hprt1* > *Bud23* > *Gapdh* > *Ywhaz* > *18S rRNA* > *Actb*.

#### 2.2.4. GeNorm

In the CA1 hippocampal region, geNorm identified the highest stability for *Actb* and *Hmbs* (stability value (M) = 0.276) and indicated *18S rRNA* as the least stable (stability value (M) = 0.658) (Figure 3D1, Appendix A). *Hprt1* and *Ywhaz*, with a stability value of 0.348, were the most stable, and *Actb*, with a stability value of 0.643, was the least stable in the CA2-3,DG sector (Figure 3D2, Appendix A). *Gapdh*, *Bud23*, and *Ywhaz* had an expression stability value (M) below the 0.5 threshold in the CA1 sector, and *Gapdh*, *Hmbs*, and *Bud23* were below the threshold in the CA2-3, DG hippocampal sector.

#### 2.2.5. RefFinder in CA1 and CA2-3,DG

The comprehensive stability ranking by RefFinder showed that *Hmbs* (1.41) and *Hprt1* (1.19) were the most stable reference genes in CA1 and CA2-3,DG, respectively (Figure 4A,B, Appendix A). The second and third most stable genes were *Gapdh* (1.86) and *Actb* (2.06) for CA1, and *Hmbs* (2.21) and *Ywhaz* (2.66) for CA2-3,DG. *Bud23* was the fourth in order of stability in both hippocampal sectors. The order of the remaining genes in CA1 was as follows: *Ywhaz* > *Hprt1* > *18S rRNA*, and in CA2-3,DG the following order was observed: *Gapdh* > *18S rRNA* > *Actb*.

### 2.3. RefFinder in Combined Hippocampal Samples

Although the two investigated regions of the hippocampus did not share the same most stable reference gene, the analysis of combined hippocampal samples was performed (Table 2, Figure 5). According to RefFinder, *Hmbs* was the most stable gene with a geometric mean of 1, which meant that *Hmbs* was ranked first in all four algorithms (Table 2). The next in order of stability was *Bud23* (2.28) followed by *Gapdh* (2.63). *Bud23* was in the first place in the geNorm analysis and the third place in the remaining algorithms. *Gapdh* was second in the delta Ct method and Normfinder, third in geNorm, and fourth in BestKeeper analysis. The least stable reference gene for CA1 + CA2-3,DG was *18S rRNA*, which was the last in four algorithms.

## 3. Discussion

This study evaluated seven candidates for the reference gene in the gerbil model of transient ischemic insult to find the most reliable normalization for qRT-PCR to further analyze changes in the expression of specific genes in search of mechanisms responsible for the differential sensitivity of the hippocampal sectors. To our knowledge, no other data have been published on this aspect in the gerbil model of transient brain ischemia, to date. Commonly used housekeeping genes, such as *Actb*, *Gapdh*, or *18S rRNA* [33], are not validated for certain experimental settings, and normalization with such genes may lead to misinterpretation of target gene expression, especially the genes with low expression. In this study, in addition to the above, we evaluated three more genes that appeared interesting as potential housekeeping genes (*Hmbs*, *Hprt1*, and *Ywhaz*). Furthermore, we selected a new gene (*Bud23*) with potentially stable expression observed in our model and reported in other models.

Our results showed that *Hmbs* and *Hprt1* were the most stable genes in the ischemia-vulnerable CA1 and ischemia-resistant CA2-3,DG sectors, respectively. For two-gene normalization in CA1, *Gapdh* and *Actb* could be used as the second gene, while *Hmbs* and *Ywhaz* could be applied in CA2-3,DG. Since the gene stability ranking differs between CA1 and CA2-3,DG, we evaluated the combined data and *Hmbs* appeared to be the most stable and suitable gene for collective normalization. Although *Bud23* was second in order of stability, we recommend using *Gapdh* as the second gene, as it displayed a higher expression level (mean Ct 21.42) than *Hmbs* and *Bud23.* Considering that the expression levels of the genes evaluated in the future experimental settings are yet to be identified, normalization with the combination of highly/medium- and lowly expressed housekeeping genes is suggested [33,34].

The *Hmbs* gene encodes the enzyme named porphobilinogen deaminase, which is the third enzyme in the heme biosynthetic pathway. The exact role of the *Hmbs* gene product in brain ischemia is unknown. It is well known, however, that heme is endogenously synthesized, also in the nervous tissue, and it regulates important biological processes such as gene transcription, microRNA processing, and signal transduction [35,36]. However, intracellular accumulation of heme and its precursors is toxic [37,38], so its synthesis and degradation are tightly regulated [39,40,41]. In CA2-3,DG *Hmbs* gene expression is less stable than in CA1. It could be explained by a post-ischemic increase in heme oxygenase-1 (HO-1), the enzyme responsible for heme degradation, in the CA2-3,DG sector of the hippocampus [42]. Additionally, it may reflect feedback from reduced free heme [43]. Nevertheless, *Hmbs* is the most stable housekeeping gene in the CA1 sector of the hippocampus also when combined data analysis is conducted. This is consistent with previous reports involving the mouse tMCAO [13], rat tMCAO [44], and rat three-vessel MCA occlusion models [45], as well as in the cardiac ischemia/reperfusion rat model [46] and canine brain [47].

Our data show that in all the tested samples, each of the four algorithms, considered separately or comprehensively, indicated *18S rRNA* or *Actb* as the genes with the least stable expression in the ischemia-resistant CA2-3,DG.

*18S rRNA* is a structural RNA and plays a role in the basic cellular process, i.e., translation. Consequently, *18S rRNA* was frequently used as a housekeeping gene. However, there is growing evidence that it is regulated by numerous biological stimuli [33]. *18S rRNA* was previously reported as an unstable housekeeping gene in the in vitro oxygen-glucose deprivation and rat pMCAO model [48], the rat ischemic wound model [48], and the rat chronic intermittent hypoxia (CIH) model [49], but as a stable one in MCAO in mice [20]. Our results show that *18S rRNA* is not suitable as a reference gene in the gerbil transient brain ischemia model. Additionally, we evaluated the *Bud23* gene, which is also associated with translation. This gene encodes protein with two independent functions, m7G methylation in 18S rRNA and ribosomal maturation [50]. There are no previously reported data concerning *Bud23* as a reference gene. Nonetheless, our results show that it is one of the most stable genes in combined hippocampal samples in the transient ischemia model in the gerbil brain.

The expression levels of *Gapdh* and *Actb* genes have been reported to change under various conditions [51,52], such as hypoxia in the mouse brain [53], hypoxia in human cell culture [54,55], and focal cerebral ischemia in rats [56,57]. While β-actin is an element of actin filaments in neurons, and these filaments are actively involved in ischemic signal transduction, it is possible that a decrease in F-actin filaments in CA1 hippocampus [58] may also influence beta-actin content and thus influence its gene expression. This was shown by in situ hybridization and by Northern blot analysis in gerbil hippocampus following 10-min cerebral ischemia [59,60]. The β-actin mRNA transiently appeared to increase after 8 h of recirculation in the stratum radiatum of the hippocampus but then declined and disappeared when CA1 neurons began to disintegrate [59]. In the same model, in Northern blot analysis, β-actin mRNA showed an increase after 6 h and 24 h of recirculation in the forebrain. There was wide variation in its expression 3 days postischemia, and by day 7 it had returned to the control value while β-actin was only faintly visible in the CA1 region at 7 days after the insult [60]. These data indicate that ischemia inhibits mRNA expressions of cytoskeletal protein in the selectively vulnerable region of the brain, i.e., CA1. In another model, the expression of β-actin increased significantly within 21 days, and that of *Gapdh* decreased at 24 h after permanent MCAO in rats, thus the authors postulated that it might not be a good housekeeping gene in less than 24-h cerebral ischemia models [21].

The empirical results reported in this paper should be considered in the light of certain limitations. The RefFinder algorithm typically uses raw Ct values without including primer efficiency, as the original geNorm and NormFinder packages do. Additionally, BestKeeper ranking in RefFinder is based on standard deviations, not the correlation coefficients of each individual gene, with the geometric mean of all genes as original software. The differences between algorithms could affect the classification of the reference gene. Therefore, the use of the RefFinder tool can lead to a suboptimal choice of a housekeeping gene, and yet, it still appears to be one of the most stable of the evaluated genes [61]. However, as shown in Figure 3, when applied separately, each algorithm provided a different representation of post-I/R stability, of the genes studied here. By selecting a reference gene based only on one of them, we may obtain unreliable results for the expression of the target genes. The data presented here are derived from potential reference gene expression analyses in different I/R-reactive areas of the hippocampus. Such genes should be included in future studies so that we could learn more about early differences in the response of neurons and identify molecular cascades that may determine their survival, which is important in the search for new therapeutic targets. However, it can often seem inappropriate to use different reference genes to study different areas in the same model, so our analyses of reference gene stability are presented here, both in two areas individually and jointly across the hippocampal formation.

## 4. Materials and Methods

### 4.1. Ethical Statement and Animals

Mongolian gerbils (*Meriones unguiculatus*) were obtained from the animal house of the Mossakowski Medical Research Institute of the Polish Academy of Sciences. Animal care was in accordance with the European Communities Council Directive (86/609/EEC). The experimental procedures were approved by the Local Commission for the Ethics of Animal Experimentation no. 2 in Warsaw (WAW2/032/2021) and every effort was made to minimize animal suffering.

### 4.2. Transient Brain Ischemia in Gerbils

Adult male gerbils weighing 60 to 70 g were subjected to transient brain ischemia by bilateral ligation of the common carotid arteries for 5 min under isoflurane anesthesia, under strictly controlled normothermic conditions, as previously described [7,8]. After ischemia, the animals recovered for 3, 48, or 72 h before decapitation, with the CA1 and CA2-3,DG regions of the hippocampus isolated for RNA. The hippocampi of sham-operated animals served as controls. The animals were randomized for the experiments.

### 4.3. RNA Isolation and Synthesis of cDNA

The fresh hippocampal sections were homogenized in Fenozol reagent and stored at −80 °C. Total RNA was purified using the Total RNA Mini Plus Concentrator Kit (A&A Biotechnology, Gdansk, Poland), according to the manufacturer’s instructions. To evaluate RNA quality and yield, the A260/A280 and A260/A230 ratios for RNA samples were analyzed with a DeNovix DS-11 FX+ spectrophotometer (DeNovix Inc., Wilmington, DE, USA). cDNA synthesis was performed using a high-capacity RNA-to-cDNA kit (ThermoFisher Scientific, Waltham, MA, USA), following the manufacturer’s instructions. 2 µg of RNA were used per 20 µL total reaction volume. Transcription was performed at 37 °C for 60 min and inactivated at 95 °C for 5 min. The cDNA was stored at −20 °C until it was used.

### 4.4. Gene Selection and Primer Design

The primers were designed using Primer3web version 4.1.0, https://primer3.ut.ee/ (accessed on 31 January 2022) [62] based on available DNA sequences of selected *Meriones unguiculatus* genes in the database of the National Center for Biotechnology Information (Bethesda, MD). Primer efficiencies were determined by construction of a standard curve using 5-fold serial dilutions of pooled cDNA template. The PCR efficiency (E) for each primer set was determined based on the slope of the standard curve with the equation: E% = (10[−1/slope] − 1) × 100%. The sequences of primers used for PCR efficiencies, and squares of the correlation coefficient (R^2^) values are listed in Table 1.

### 4.5. qRT-PCR

For qRT-PCR, PowerUp Sybr Green Master mix (ThermoFisher Scientific) was used following the manufacturer’s instructions with primer concentrations of 400 nM and 100 ng of cDNA per reaction. qRT-PCR was carried out on the Applied Biosystems 7500 Fast Real-Time PCR System in triplicates. The PCR conditions included an UDG activation step at 50 °C for 2 min and a polymerase activation step at 95 °C for 2 min, followed by 40 denaturation cycles at 95 °C for 3 s and annealing/extension at 60 °C for 30 s. To determine the specificity of the reaction products, a melting curve analysis was performed, as follows: samples were ramped to 95 °C at a rate of 1.6 °C/s, cooled to 65 °C for a period of 1 min, and then slowly heated to 95 °C at a rate of 0.15 °C/s. The cycle threshold values and the melting curve were analyzed with the SDS 2.3 software (Applied Biosystems, Waltham, MA, USA).

### 4.6. Statistical Data Analysis

The mean Ct value was calculated from three technical replicates using the SDS 2.3 software and was taken to be the Ct that represents the biological sample. Ct data was exported to Microsoft Excel to calculate the arithmetic mean and standard deviation for each gene. Next, Ct data were exported to GraphPad Prism version 9.4.1 (GraphPad Software, San Diego, CA, USA) to calculate median, 25th and 75th percentiles, maximum and minimum values, and to generate box plot and column graphs.

### 4.7. Data Analysis of Gene Expression Stability

qRT-PCR data of candidate genes were analyzed for expression stability using the online tool RefFinder, https://www.heartcure.com.au/reffinder/ (accessed on 13 April 2022) [32]. The tool integrates four major computational algorithms: delta Ct, Bestkeeper, NormFinder, and geNorm, and allows for the comparison and ranking of experimental candidates. Based on the rankings of each algorithm, the tool calculates geometric means in order to create a final overall ranking—comprehensive gene stability. The calculation included Ct values of samples from each time point (control, 3, 48, and 72 h) and both regions of the hippocampus in 4 biological replicates. Both regions, CA1 (n = 15) and CA2-3,DG (n = 15), were analyzed separately and in combination.

#### 4.7.1. Delta Ct (ΔCt) Method

The delta Ct method is used to compare the relative expression of pairs of genes within each sample, and the average SD allows for the genes to be ranked. The most stable gene has the lowest average SD [63].

#### 4.7.2. BestKeeper

For Bestkeeper, the standard deviation and coefficient of variation (CV) are used to calculate the stability of the expression of candidate reference genes with raw Ct values, and the highest stability is represented by the lowest CV [64].

#### 4.7.3. NormFinder

For NormFinder, an ANOVA-based model of each reference gene is used to calculate the expression stability value based on intra- and intergroup variations. The most stable gene has the lowest stability value [65].

#### 4.7.4. GeNorm

GeNorm analysis calculates the average pairwise variation (V) between a particular gene and all other control genes. The gene with the lowest average expression stability value (M) is the most stable [66].

## 5. Conclusions

To conclude, this study provides information on the stability and selection of the reference genes for qRT–PCR normalization in the hippocampus of gerbils subjected to transient ligation of the common carotid arteries. The obtained results indicate that *Hmbs* is the most reliable housekeeping gene in the whole hippocampal formation during postischemic reperfusion. Since *Bud23*, the second gene in the post-I/R expression stability ranking, has a comparably low level of expression as *Hmbs*, we recommend *Gapdh* be used as a second reference gene due to its higher expression level.

## Figures and Tables

**Figure 1 ijms-24-02756-f001:**
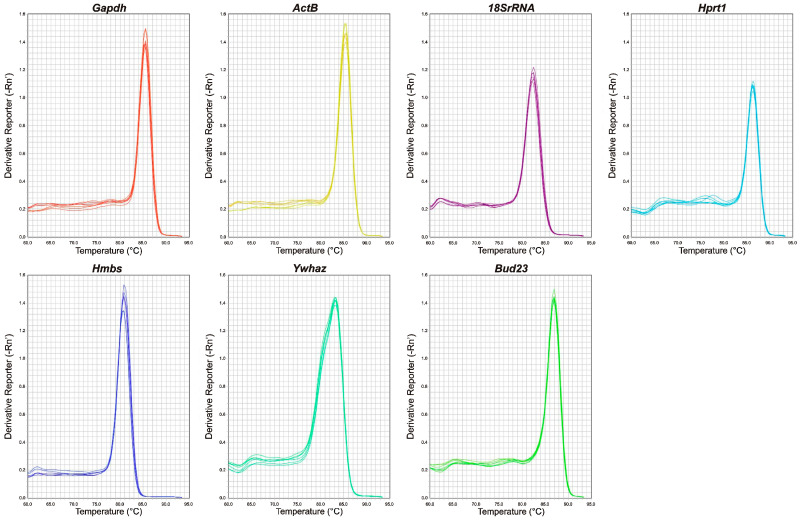
Validation of qRT-PCR primers. Melting curve analysis for testing the specificity of gerbil primers for candidate reference genes. A single peak indicates a single PCR product.

**Figure 2 ijms-24-02756-f002:**
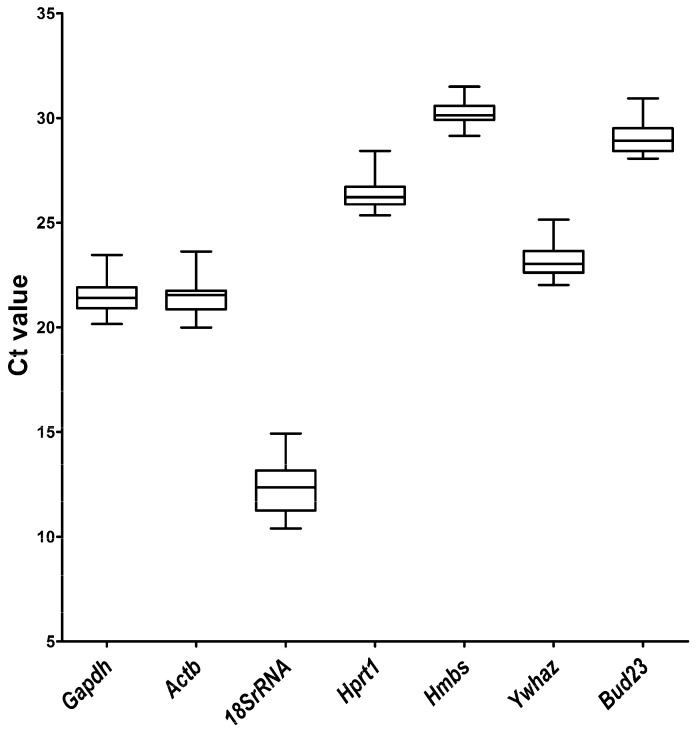
Distribution of the qRT-PCR cycle threshold (Ct) values for the candidate reference genes in all experimental groups of gerbil hippocampi (n = 30). A line across the box is depicted as the median. The boxes indicate the 25th and 75th percentiles. Whiskers represent the maximum and minimum values.

**Figure 3 ijms-24-02756-f003:**
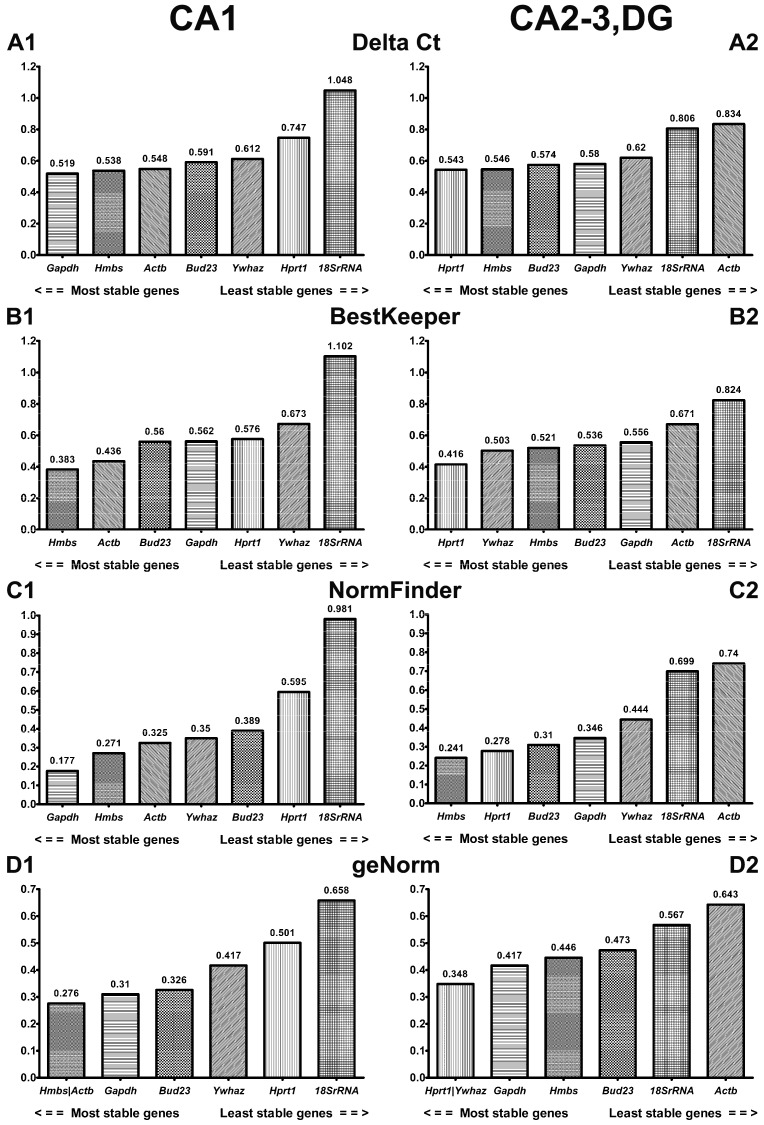
Stability rankings of seven candidate reference genes in CA1 and CA2-3,DG parts of gerbil hippocampus, calculated separately by four algorithms: ΔCt, BestKeeper, NormFinder, and geNorm. Data are presented from the most stable (left) to the least stable gene. n = 15. (**A1**). ΔCt in CA1; (**A2**). ΔCt in CA2-3,DG; (**B1**). BestKeeper in CA1; (**B2**). BestKeeper in CA2-3,DG; (**C1**)**.** NormFinder in CA1; (**C2**). NormFinder in CA2-3,DG; (**D1**). geNorm in CA1; (**D2**). geNorm in CA2-3,DG.

**Figure 4 ijms-24-02756-f004:**
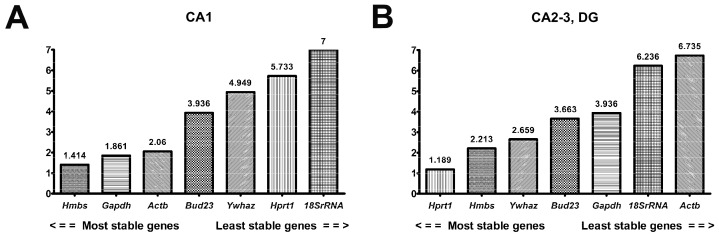
RefFinder stability rankings of 7 candidate reference genes in CA1 and CA2-3,DG parts of hippocampus, calculated separately (**A**,**B**). Data are presented from the most stable (left) to the least stable gene. n = 15.

**Figure 5 ijms-24-02756-f005:**
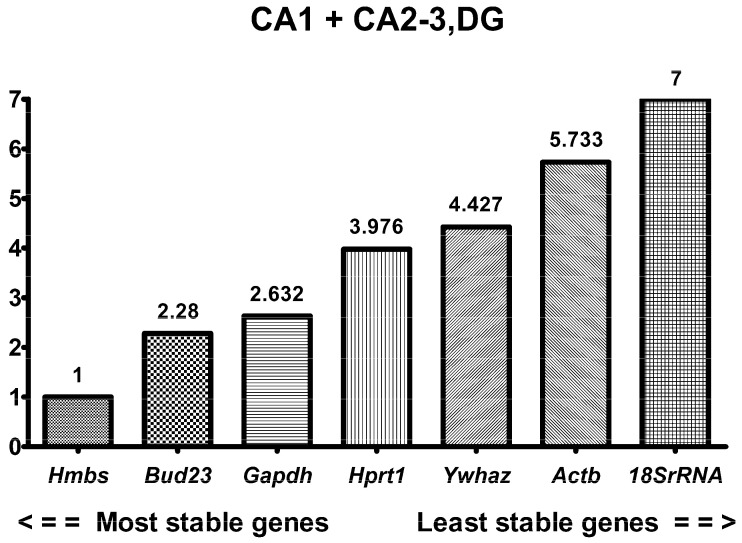
RefFinder stability rankings of 7 candidate reference genes in CA1 + CA2-3,DG parts of gerbil hippocampus, calculated collectively. Data are presented from the most stable (left) to the least stable gene. n = 30.

**Table 1 ijms-24-02756-t001:** Mongolian gerbil primer sequences used in this study for quantitative polymerase chain reaction (qRT-PCR) and efficiency of chain polymerization reactions. R^2^—regression coefficient.

Gene	Accession Number	Primer Sequence (5′–3′)	Amplicon Size (bp)	Primer Efficiency	R^2^	Reference
*Gapdh*	XM_021636934	F: AGTATGACTCTACCCACGGC	150	103.94%	0.999	[30]
		R: ACTCCACAACATACTCGGCA				
*Actb*	XM_021652973	F: CTCTGGCTCCTAGCACCATG	129	96.39%	0.999	
		R: ACTCCTGCTTGCTGATCCAC				
*18S rRNA*	AJ877917	F: GGCTACCACATCCAAGGAAGG	101	99.67%	0.999	[31]
		R: AGGGCCTCGAAAGAGTCCTG				
*Hprt1*	XM_021660111	F: TTACGGCTTTCCTGGAGGTG	136	90.10%	0.999	
		R: TAGCCTGGTTCATCATCGCC				
*Hmbs*	XM_021659401	F: GAAGAGTGGCCCAGCTACAG	108	103.57%	0.999	
		R: CACTGAACTCCTGCTGCTCA				
*Ywhaz*	XM_021657529	F: CACTCACTCCGGACACAGAA	197	107.05%	0.997	
		R: CCTACGGGCTCCTACAACAT				
*Bud23*	XM_021664516	F: GTGGCTCTGCAATGCAAACA	132	104.37%	0.998	
		R: TGCTCCGAGTTCTCAGGGTA				

**Table 2 ijms-24-02756-t002:** Stability value of reference gene from RefFinder and each algorithm. The term *Combined* represents the results of analysis for all samples, from CA1 and CA2-3,DG regions of gerbil hippocampus. SV and R stand for stability value and rank for each method, respectively.

Region of Hippocampus	Gene Name	RefFinder		Delta Ct		BestKeeper		NormFinder		geNorm	
		SV	R	SV	R	SV	R	SV	R	SV	R
Combined	*Gapdh*	2.632	3	0.598	2	0.575	4	0.319	2	0.455	3
	*Actb*	5.733	6	0.741	6	0.593	5	0.585	6	0.57	6
	*18S rRNA*	7.00	7	1.018	7	0.962	7	0.936	7	0.698	7
	*Hprt1*	3.976	4	0.67	5	0.525	2	0.447	5	0.529	5
	*Hmbs*	1.00	1	0.59	1	0.452	1	0.26	1	0.422	1
	*Ywhaz*	4.427	5	0.656	4	0.647	6	0.432	4	0.507	4
	*Bud23*	2.28	2	0.612	3	0.534	3	0.341	3	0.422	1

## Data Availability

Data is contained within this article and Appendix A.

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
