# Peer review of "Validation of the Reference Genes for Expression Analysis in the Hippocampus after Transient Ischemia/Reperfusion Injury in Gerbil Brain"

_ijms, 2023, doi:10.3390/ijms24032756_

Round 1

Reviewer 1 Report

The present manuscript entitled “Validation of the Reference Genes for Expression Analysis in the Hippocampus after Transient Ischemia/Reperfusion Injury in Gerbil Brain” demonstrated the candidates for housekeeping genes in the hippocampal formation using a gerbil model of transient forebrain ischemia. The present results suggest that their findings provide fundamental data to deeply and integratively comprehend selective vulnerability according to brain regions following ischemia-reperfusion injury. In addition, the authors employed a suitable approach to normalize the housekeeping genes. However, this reviewer noticed some problems which are needed to be revised for potential publication. The points have been summarized as below: 

1. The authors should clearly and separately describe their findings and suggestion in the “Abstract”.

2. There is lack of description for some abbreviations such as CA and DG in the "Abstract" section and MCAO in the "Introduction" section. Please specify these abbreviations in the whole-text.

3. In the last paragraph of the "Introduction" section, the results of the present study are described; it is inappropriate that. Instead, the purpose of the present study should be described.

4. The authors investigated the housekeeping genes at three, 48 and 72 hours after ischemia/reperfusion (IR). However, it is well acknowledged that loss/death of pyramidal neurons in the hippocampal CA1 region occurs at four to five days after IR induced by five minutes transient forebrain ischemia in gerbils. In this regard, the authors have to investigate the housekeeping genes at four or five days after IR: the authors need to add the data.

5. Results displayed by the authors are hardly distinguishable. Please provide higher quality of figures: the letters are hard to be read. Furthermore, the present data showing the stability rankings of seven candidate reference genes were presented according to the hippocampal regions. Instead, the results should be displayed according to the experimental groups (sham-operated group and ischemia-operated group), and points in time after IR (three, 48 and 72 hours after IR).

6. The authors investigated the housekeeping genes in two regions of the hippocampal formation: 1) CA1 region, and 2) CA2/3 and DG. However, the principal cells located in the CA1-3 regions (pyramidal neurons) differ from those in the DG (granular cells). Therefore, the authors should divide the hippocampal formation into three regions: 1) CA1 region (pyramidal neurons which are vulnerable to IR injury), 2) CA2/3 region (pyramidal neurons which are tolerant to IR injury), and 3) DG (granular cells which are more tolerant to IR injury).

7. In conclusion, the authors only indicated their data. The authors should describe a brief result and indicate their findings. In addition, it is good to suggest their study for clinical applications.

8. This reviewer noticed that the language style of the present manuscript should be entirely revised by a native speaker.

Reviewer 2 Report

The authors presented data for the use of reference genes in global ischemia in the gerbil model in CA1 and CA2-3 of the dendate gyrus. Design and methods are sound and presented transparently. The need is obvious since shifts in the reference genes introduces a bias in all data of animals subjected to ischemia. 

Author Response

We are thankful for all the Reviewers’ comments. We appreciate the time and effort the Reviewers dedicated to providing feedback on our manuscript.

Reviewer 3 Report

This article provides relevant information that improves the methods used in the evaluation of gene expression in a model of cerebral ischemia. 

so I recommend its publication

Author Response

(The authors gave the same response as above.)

Round 2

Reviewer 1 Report

This reviewer consider the manuscript can be accepted in present form.